# The Role of Non-Coding RNAs in the Pathogenesis of Parkinson’s Disease: Recent Advancement

**DOI:** 10.3390/ph15070811

**Published:** 2022-06-30

**Authors:** Hanwen Zhang, Longping Yao, Zijian Zheng, Sumeyye Koc, Guohui Lu

**Affiliations:** 1Department of Neurosurgery, First Affiliated Hospital of Nanchang University, Nanchang 330006, China; helenzhung1210@163.com (H.Z.); zijian.zheng@163.com (Z.Z.); 2The Center of Neuroscience, Life Science Zurich Graduate School, University of Zurich, 8091 Zurich, Switzerland; 3Department of Neuroscience, Institute of Health Sciences, Ondokuz Mayıs University, Samsun 55200, Turkey; sumeyyekoc0234@gmail.com

**Keywords:** Parkinson’s disease, non-coding RNAs, neuroinflammation, microglia, dopaminergic neurons

## Abstract

Parkinson’s disease (PD) is a prevalent neurodegenerative aging disorder that manifests as motor and non-motor symptoms, and its etiopathogenesis is influenced by non-coding RNAs (ncRNAs). Signal pathway and gene sequence studies have proposed that alteration of ncRNAs is relevant to the occurrence and development of PD. Furthermore, many studies on brain tissues and body fluids from patients with PD indicate that variations in ncRNAs and their target genes could trigger or exacerbate neurodegenerative pathogenesis and serve as potential non-invasive biomarkers of PD. Numerous ncRNAs have been considered regulators of apoptosis, α-syn misfolding and aggregation, mitochondrial dysfunction, autophagy, and neuroinflammation in PD etiology, and evidence is mounting for the determination of the role of competing endogenous RNA (ceRNA) mechanisms in disease development. In this review, we discuss the current knowledge regarding the regulation and function of ncRNAs as well as ceRNA networks in PD pathogenesis, focusing on microRNAs, long ncRNAs, and circular RNAs to increase the understanding of the disease and propose potential target identification and treatment in the early stages of PD.

## 1. Introduction

Non-coding RNAs (ncRNAs) are a cluster of unique transcripts that regulate cell function through various mechanisms that control gene expression at the transcriptional and post-transcriptional levels [1,2,3,4]. Although they are incapable of being transcribed as proteins to constitutive cell structures or modulate physiological processes, their specific regulatory mechanisms, such as competing for endogenous RNAs (ceRNAs), also are involved in the development of many diseases. It is widely acknowledged that some ncRNAs are closely associated with neurodegenerative diseases owing to their massive expression of transcripts and potential signaling pathways in the central nervous system (CNS), such as the substantia nigra striatum [5,6,7]. Notwithstanding such discoveries in ncRNAs, our understanding of the connection between ncRNAs and Parkinson’s disease (PD) pathogenesis is largely superficial. One possible mechanism is that ncRNAs participate in neuroinflammation mediated by microglial cells [8,9,10,11,12]. In recent years, many pathogenic factors have been associated with neurodegeneration in nigrostriatal dopamine neuron apoptosis or death caused by neuroinflammation [13,14,15,16] (Figure 1). These factors are potential targets that interfere with the disease process of PD [8,17,18]. In this review, we primarily concentrate on miRNAs, lncRNAs, and circRNAs, as well as the details of their roles and potential regulatory mechanisms in the pathogenesis of PD.

## 2. ncRNAs

### 2.1. Introduction

Non-coding (ncRNAs) RNAs are broadly defined as a cluster of RNA transcripts that are incompetent to code proteins. In recent years, as the development of sequence and structure analysis has accelerated, techniques such as next-generation sequencing have deepened our understanding of ncRNAs. Most genomes can be converted into RNAs. However, only 2% of that can ultimately be translated into proteins. Thus, RNAs are classified into two types, coding RNA and ncRNAs; ncRNAs are further divided into structural and regulatory ncRNAs. From an evolutionary perspective, structural ncRNAs are conserved and expressed in almost all species of creatures, including plants, yeast, viral and procaryotic organisms, and the housekeeping genes for these creatures are rRNA and tRNA. Concurrently, the conservation of regulatory ncRNAs is poorer than that of structural ncRNAs, including small ncRNAs such as miRNAs, medium-long ncRNAs, and long non-coding RNAs (lncRNAs). This observation is consistent with the conclusion that ncRNA explains the complexity of multicellular organisms. Although ncRNA was initially recognized as “dark transcriptome” or “genomic dark matter” [19,20], an increasing number of reports has demonstrated that ncRNA exhibits a crucial role in regulating cellular pathways and functions on both genetic and epigenetic levels. For instance, ncRNAs can guide DNA synthesis and genome rearrangement and protect genes from exogenous nucleic acids [3]. The mechanism of ncRNAs has been a topic of interest in recent years. Some underlying mechanisms, such as competing endogenous RNA (ceRNA), have been confirmed, but many potential mechanisms are not fully understood.

### 2.2. ncRNAs in PD

The brain contains several types of ncRNAs, including miRNA, lncRNA, circRNA, and piwiRNA, which play essential roles in neuronal growth, differentiation, organization, operation, and protection of the CNS throughout the life cycle. Based on fundamental experimental studies on the potential mechanisms, insights from such research offer substantial proof that ncRNAs can regulate diverse signaling pathways associated with neurodegeneration, including cell apoptosis, mitochondrial dysfunction, oxidative stress, altered protein handling, neuroinflammation, and specific protein aggregation [21,22,23,24]. Some lncRNA, miRNA, and circRNA can also participate at the transcriptional and post-transcriptional levels through the ceRNA pathway to regulate target proteins and thus affect neurodegeneration. Others can directly regulate related proteins, and exosomes secreted by neurons can control some proteins. Although the molecular mechanism of PD has been thoroughly researched, the occurrence and promotion mechanisms of PD remain unclear.

## 3. LncRNA

### 3.1. Introductionn

Long non-coding RNA (lncRNA) refers to enormous RNA families defined by a length over 200 nucleotides, limited protein-coding potential, and no detectable open reading frames, which are necessary for protein-coding potential. Meanwhile, lncRNAs are considered to serve as a cryptic, but critical, layer in the genetic regulatory code associated with diverse physiological and pathological responses, dysfunction is closely related to the occurrence of diseases [25].

### 3.2. lncRNAs in the Pathogenesis of PD

lncRNAs have received increased research focus, especially regarding neurodegeneration in brain function and CNS disorders. Many lncRNAs have been reported to be abnormally expressed in the cytoplasm in PD. For example, Kraus performed a comprehensive analysis of the expression levels of 90 well-annotated lncRNAs in 30 brain specimens derived from 20 patients with PD and 10 controls as a preliminary report on the significance of lncRNAs in PD. They found that the H19 upstream conserved 1 and 2 is significantly downregulated in PD. Additionally, lincRNA-p21, LINC-PINT, metastasis-associated lung adenocarcinoma transcript 1 (Malat1), small nucleolar RNA host gene 1 (SNHG1), and tiny non-coding RNAs are significantly upregulated [26]. Additionally, microarray analysis has reported that the upregulation of lncRNAs HOTAIRM1 and AC1131056.3 and downregulation of lncRNA XIST may contribute to PD pathogenesis through dopaminergic neuron injury [23,27]. M’arki et al. predicted that lncRNA BCYRN1 SNPs rs13388259 is associated with PD because it is located close to the binding site of the transcription factor HNF4A, which is upregulated in peripheral blood [28].

Additionally, abnormal deposition of α-syn has been implicated in the function of lncRNAs. lncRNA NEAT1, previously known as an oncogene in malignant tumors, was found to accelerate α-syn aggregation and simultaneously promote neuronal apoptosis in MPTP-induced PD mice [29]. Lin et al. found that lncRNA G069488, RP11-142J21.2 and AC009365.4. corresponded to α-syn deposition in α-syn oligomer-induced cells using microarray analysis [21].

Autophagy plays a significant role in PD pathogenesis. The disharmony of autophagy is significantly associated with α-syn aggregation. L1CAM, an exosome derived from neurons, has been reported to correspond to the autophagic-lysosomal pathway, at least partly participating in autophagy. Zou et al. demonstrated that linc-PIU3F3 exerted a positive regulatory effect on α-syn in L1CAM [30]. Similarly, the α-syn pathogenetic aggregation, lncRNA-T199678 has been identified to serve as a sponge of miR-101-3p to relieve dopaminergic neuron injury [31]. Additionally, Huang et al. and Quan et al. recruited patients with PD and healthy controls to evaluate the expression of lncRNA MEG3, which is downregulated in patients with PD. They found that overexpression of MEG3 protected MPP+ SH-SY5Y cells from apoptosis by boosting LRRK2 expression [32,33].

Another pathogenesis in which lncRNAs participate is that of neuroinflammation. Together, lincRNA-p21 and miR-181/PKC-δ forms a double-negative feedback loop that facilitates sustained microglial activation and the deterioration of PD [34]. A recent study showed that overexpression of lncRNAGAS5-activated microglia promoted NLRP3 expression and consequently improved neuroinflammation in PD by sponging the miR-223-3p/NLRP3 signaling pathway [35].

Abnormal expression of lncRNAs in the cytoplasm has been shown to be associated with apoptosis. Lnc-MKRN2-42:1 in exosomes has been reported to contribute to the pathogenesis of PD using next-generation sequencing and bioinformatics analysis [36]. LINC-PINT, a cancer-promoting gene, has recently been found to exert a neuroprotective role in PD [37], and lncRNA GAS5 is expressed differently in the hippocampus region of young and old mice and is associated with cell proliferation and apoptosis [38]. Additionally, lncRNA UCA1 and lncRNA HOTAIR promote caspase-3 activation and enhance apoptosis in the MPP+-induced cell model by respectively upregulating the expression of SNCA and LRRK2 [39,40] (Table 1) (Figure 2).

### 3.3. lncRNAs in the Neuroinflammation of PD

The remaining abnormally labeled lncRNAs have been demonstrated to mediate inflammatory responses in other diseases, but they seem to be under investigation in the inflammatory pathogenesis of PD. SNHG1 promotes neuroinflammation in the pathogenesis of PD by modulating the miR-7/inflammasome pyrin domain-containing 3 (NLRP3) pathway. Downregulation of SNHG1 suppresses the activation of microglia and NLRP3 inflammasome, as well as DA neuron loss in the midbrain substantia nigra, pars compacta (SNpc) in MPTP-induced PD mice [43]. Additionally, Wang et al. Found that SNHG1 facilitates neuronal injury by modulating miR-181a-5p/CXCL12, miR-125B-5p/MAPK1, miR-216A-3P/Bcl-2, miR-221/222/p27/mTOR, miR-15b-5p/GSK3β, and the miR-153-3p/PTEN/AKT/mTOR axis in neuroblastoma cells [22,44,45,46,47,48,49]. HOXA-AS2 was considerably upregulated in peripheral blood mononuclear cells of patients with PD and showed a reversible regulatory relationship with PGC-1α expression. HOXA-AS2 promotes neuroinflammation by accelerating the conversion of microglia to an anti-inflammatory M2 phenotype [41].

Moreover, the lncRNA HOXA11-AS was overexpressed in an animal model of PD: MPTP (+)-induced NLRP3 inflammatory expression and LPS-induced microglial activation in a PD cell model was enhanced by HOXA11-AS and FSTL1 upregulation and/or miR-124-3p knockdown [50]. Moreover, activating the inflammasome and producing ROS are important mechanisms by which MALAT1 is involved in the pathogenesis of PD by restraining Nrf2 expression in both microglial cells and mouse models [42]. Inhibition of lncRNA RMST mitigated oxidative stress and neuronal apoptosis, augmented the expression of TH and SYN, and induced neuronal impairment through the TLR/NF-κB signaling pathway [51]. miR-625/TRPM2 is another neuroinflammation-related signaling pathway that promotes oxidative stress via lnc-p21, leading to PD development [52] (Table 1) (Figure 2).

### 3.4. Circulating lncRNA in the Pathogenesis and Neuroinflammation of PD

Circulating lncRNAs have been recently reported to be dysregulated in the circulation of patients with PD, which indicates that the expression level of circulating lncRNAs might be considered a biomarker for PD. Kasra et al. profiled lncRNA expression in the peripheral blood of patients with PD and healthy controls using qRT-PCR. HULC, PVT1, MEG3, SPRY4-IT1, LINC-ROR, and DSCAM-AS1 lncRNA expression levels were measured. HULC and PVTA levels were lower in patients with PD, while MEG3, SPRY4-IT1, LINC-ROR, and DSCAM-AS1 levels were higher. HULC and PVTA have been shown to regulate apoptosis [53,54,55]. These results indicate the potential role of circulating lncRNAs as treatment markers for PD. However, none of these lncRNA expressions were related to patient age, disease stage, Mini-Mental State Examination scores, or Unified Parkinson’s Disease Rating Scale scores [56].

Previous studies have shown that nuclear factor-κB (NF-kB) regulates the activity of inflammatory intermediates during inflammation and oxidative stress, which contribute to the neurodegenerative process in PD [57,58]. Thus, to appraise their possible application as disease markers, Ghafouri-Fard et al. identified circulating lncRNAs involved in the modulation of NF-kB signaling in the circulation of patients with PD versus healthy individuals, including CEBPA, ATG5, PACER, DILC, NKILA, ADINR, DICER1-AS1, HNF1A-AS1, CHAST, and H19. PACER and HNFA1-AS were lower, while DILC, CEBPA, ATG5, and H19 were higher. CEBPA, DILC, and ATG5 were determined to be the most appropriate markers. These results provide evidence for the participation of NF-κB-related lncRNAs in the pathogenesis of PD [59].

## 4. microRNA (miRNA)

### 4.1. Introduction

miRNAs are small ncRNA molecules containing approximately 22 nucleotides that are initiated from pre-miRNAs transcribed by RNA pol II [60,61,62,63] (Figure 2). It could play a complex role in regulating transcription of multiple genes via combining with Argonaut and other proteins into the RNA-Silencing Complex. This complex binds to mRNA and promotes post-transcriptional regulation by promoting the degradation of mRNA and subsequently inhibiting the expression of target genes [64,65]. miRNAs are well-known for cancer pathogenesis. However, many miRNAs are also pivotal regulators of normal CNS function by regulating neuronal proliferation and differentiation [66,67]. Additionally, the altered expression of miRNAs could contribute to neurodegeneration in neurological disease [68], including Alzheimer’s disease [69], amyotrophic lateral sclerosis [70], and frontotemporal dementia [71]. Therefore, identifying the impact of miRNA on PD is necessary (Figure 3).

### 4.2. microRNA in the Pathogenesis of PD

The abnormal expression of miRNAs is attributable to the development and progression of numerous diseases. Similarly, in patients with PD, the serum and striatal brain tissue and DA neurons from the SNpc demonstrated dysregulation of miRNA expression. On the one hand, some miRNAs were downregulated in PD; for instance, miR-153 and miR-223 were observably decreased in the saliva, serum, and brain of patients with PD compared to normal controls [72]. An exciting result showed that the expression levels of miR-132-3p, miR-146-5p, and miR-29b were significantly decreased in the serum of patients with severe PD compared to those in regular patients with PD and the healthy group, which indicated that the levels of these miRNAs were related to memory performance and cognition [73,74]. In addition, the levels of other miRs were improved in PD, and the level of miR-132 was elevated in the PD serum compared to that in healthy controls [75]. One investigation found that miR-486-5P expression was significantly enriched in colonic biopsies from patients with PD compared to that in the control group, offering a potential gene treatment target for PD [76]. Besides miR-132 and miR-486-5P, miR-105-5p and miR-195 are similarly expressed at higher levels in idiopathic PD than in healthy groups [65,77]. miR-29c expression is notably increased in Turkish patients with PD [78]. Dysregulated miRNA expression profiles indicate potential therapeutic targets for PD. In addition to dysregulated miRNAs in the cytoplasm, an exploratory study showed that miR-34a-5p expression was markedly upregulated in small extracellular vesicles from patients with PD compared to that in normal subjects [79].

As miRNA dysregulation may participate in PD pathogenesis, we conclude the specific mechanism of miRNAs in reference to PD. Excessive activation leads to autophagy failure, thereby increasing the aggregation of α-syn. MiR-599 was found to serve as a negative regulator of LRRK2, and when microRNA-599 is expressed, LERK2 is highly expressed in cell and animal models [80]. In addition, MiR-155-5p has been reported to be expressed heavily in patients with PD [81] and modulating α-synuclein-triggered inflammatory response in a PD mice model [82]. This result was also found in PD mice treated with RA, a neuroprotective substance in PD [83]. Meanwhile, miR-153/miR-223 was shown to be downregulated by overexpression of HO-1 in stressed astroglia to promote α-syn production and toxicity [84]. In addition, upregulation of miR-204-5p also contributes to the increase in α-syn and other neurotoxicity protein aggregation via inhibition of the DYRK1A-mediated ER stress and apoptosis cascade [85].

Research findings have also shown that competitive endogenous RNAs (ceRNAs) networks regulate mitochondrial dysfunction and lead to loss of dopaminergic neurons, inducing loss of dopaminergic neurons. Failure to eliminate damaged mitochondria leads to the accumulation of dysfunctional mitochondria and DA neuronal impairment [24,86], which was verified to partially correspond to the increased amount of miR-146a bound to NF-kβ via repression of PRKN expression [87]. Several miRNAs have been demonstrated to target PINK1S, which is vital for mitochondrial autophagy, such as miR-27a and miR-27b. Kim et al. found that miR-27a/27b can bind to the 3′ untranslated region (3′UTR) of PINK1 mRNA to inhibit PINK1 function and mitophagy. Predominantly, miRNA-144 induces microglial autophagy and inflammation following intracerebral hemorrhage [88]. In contrast, miR-144 and its target gene β-amyloid precursor protein regulate MPTP-induced mitochondrial dysfunction [89].

Injured autophagy in dopamine neurons leads to α-syn in PD. miR-128 was previously predicted to be linked to TFEB, a critical gene involved in autophagy and lysosome phagocytosis, and its overexpression could rescue α-syn toxicity in midbrain neurons [90]. In brain autopsies of patients with PD, the TFEB content was highly decreased in the midbrain. The downregulation of miR-29c-3p has been reported in the SNpc DA neurons of PD mice and may be accompanied by autophagy by inhibiting TET2 [91]. Lack of IFNB/interferon-β blocked neuronal autophagy associated with impaired autophagosome degradation, leading to the accumulation of various neurotoxic aggregation-prone proteins, and an exciting study observed that miR-1 could play a pivotal role in the loss of IFNB-induced impaired autophagy by targeting TBC1D15/RAB7 [92,93]. By inhibiting the expression of XBP1, miR-326 overexpression enhances autophagy of neurons and suppresses iNOS immune-positive cells through the JNK signaling pathway in in vitro and in vivo PD models [94]. In addition to inhibiting autophagy, miR-326 suppresses apoptosis and improves the proliferation of dopaminergic neurons by repressing the MAPK/KLK7 signaling pathway [95].

Apoptosis is the terminal outcome of dopaminergic neurons in patients with PD. The mechanisms described above ultimately contribute to apoptosis. MiR-380-3p suppressed cell proliferation and aggravates PQ-induced PD, and the potential tool is that miR-380-3p block the translation of Sp3 mRNA [96]. The upregulation of miR-15b-5p represses the rate of apoptosis and caspase-3 activity in the MPP (+)-induced SH-SY5Y cell mode l and the MPTP-induced mouse model by downregulating the Akt3-mediated GSK-3β/β-catenin signaling pathway [97]. miR-185 was downregulated in a PD model, and its restoration mitigated oxidative stress by targeting IGF1 to invoke the PI3K/AKT signaling pathway [98].

In addition to experimental validation, several studies have conducted computational prediction to propose evidence of the participation of miRNAs in PD pathogenesis; ISG15, RRM2, FBXW11, FOXM1, and the miR-181 family were found to be aberrantly expressed in PD [99]. Hsa-miR-626 was significantly decreased in the cerebrospinal fluid of patients with PD [100]. Another microarray analysis showed that hsa-miR-19b-3p levels were higher than healthy [101]. Additionally, miR-7-5p, miR-331-5p, miR-145-5p, hsa-miR-335-5p/hsa-miR-3613-3p, hsa-miR-335-5p/hsa-miR-6865-3p, and miR-335-5p/miR-3613-3p/miR-6865-3p were verified to have diagnostic value for PD using other microarray analyses [102,103]. Although many miRNAs have been found to increase or decrease in patients with PD, the specific mechanism of action has not yet been elucidated. However, some studies have identified new potential therapeutic targets [104] (Table 2) (Figure 4).

### 4.3. microRNA in Neuroinflammation of PD

Recent research has shown that miRNA can directly regulate the activation of microglia and affect neuroinflammation development in PD. Some miRNAs are specifically expressed in microglia (e.g., miR-125b-5p, miR-342-3p, and miR-99a) [125]. Postmortem analysis of patients with PD and healthy subjects showed that miRNAs were negatively correlated with cytokines. The downregulation of miR-218, miR-124, and miR-144 is related to PD through the activation of the NF-κB signaling pathway [126]. Our previous study showed that miR-124 can inhibit neuroinflammation during the development of PD by regulating the mitogen-activated protein kinase kinase kinase (MEKK3)/NF-κB signaling pathways. Overexpression of miR-124 or knockdown of MEKK3 could prevent neuronal death and apoptosis following microglial activation in the microglial culture supernatant transfer model or PD mouse model [8]. We also found that miR-124 suppressed the secretion of pro-inflammatory mediators by targeting 62/p38 expression and promoting autophagy in the inflammatory pathogenesis of PD [110]. Increased miR-124 expression can suppress neuroinflammation and autophagy in PD development by inhibiting the Hedgehog signaling pathway [111]. MiR-155 regulates alpha-synuclein-induced inflammatory responses in PD models. miR-155 likely lies upstream of the primary histocompatibility complex class II (MHC II) antigen presentation because knocking out miR-155 can prevent the induction of MHC II expression in PD [82]. The change in TNF-α expression precedes the change in IL-1β, IL-6, iNOS, and COX-2 expression, and the downregulation of miR-7116-5p in microglia MPP+ sensitizes TNF-α production to induce DA neuron damage in PD [127]. MiR-7 targets nod-like receptor protein three inflammasomes to modulate neuroinflammation and protect DA neurons against degeneration in the pathogenesis of PD [128]. CDC42, a candidate gene for PD involved in neuronal death, has been identified as a potential target for miRNAs 29a-3p and 103a-3p, while Rho-associated coil-containing protein kinase (ROCK)/CDC42-mediated microglial motility and glimpse formation leads to phagocytosis of degenerating DA neurons in vivo [129]. MiR-135b upregulates cell proliferation, apoptosis, and production of inflammatory cytokines such as TNF-α and IL-1β in MPP+-intoxicated SH-SY5Y cells by directly targeting glycogen synthase kinase-3β (GSK3β) [130]. Increased miR-129 levels prevent inflammation-induced neuronal and blood–spinal cord barrier (BCSB) damage by inhibiting high-mobility group box-1 (HMGB1) and TLR3-related cytokines, thereby preventing ischemia-reperfusion (IR) [131]. By targeting AKT3, miR-150 suppresses the release of pro-inflammatory cytokines (IL-1β, IL-6, and TNF-α) and inhibits the neuroinflammatory responses of cells treated with LPS [132,133]. miR-217/miR-138-5p downregulation evokes inhibition of ROS and TNF-α release, lactate dehydrogenase (LDH) activity, and cell apoptosis [134]. MiR-375 has been shown to be observably downregulated in PD [135]. Overexpression of miR-375 was verified to ameliorate abnormal behavior and the loss of dopaminergic neurons and abate oxidative stress and release of TNF-α, IL-6, and other neuroinflammation factors via inhibition of SP1, a transcriptional factor that lowers expression, indicating favorable neuronal survival [136]. Downregulation of miR-21 promotes the release of pro-inflammatory cytokines (IL-6, IL-1β, and TNF-α) and ROS production by negatively modulating the expression of Bcl-2 [137]. Additionally, miR-3473b modulates the secretion of TNF-α, IL-β, and other inflammatory factors by targeting TREM2/ULK1 expression to affect autophagy’s function in neuroinflammation in PD [138].

In the past few years, the mechanism of miRNA and microglial activation has been further revealed, and these miRNAs also play a critical role in the pathogenesis of PD. For instance, inhibition of miR-155 modulates endotoxin tolerance by upregulating suppressor of cytokine signaling-1 in microglia. Notably, the expression of miR-155 can be modified by levodopa treatment; in fact, a downregulation of miR-155 with the highest dosage is observed in patients with PD [81,139]. Moreover, knockdown of miR-146a can suppress the activation of microglia in mice, and analysis of miRNA expression shows that miR-146a-5p expression is downregulated in patients with PD compared to that in HCs [81,140]. Accordingly, miR-7 alleviates the secondary inflammatory response of microglia caused by cerebral hemorrhage by inhibiting toll-like receptor 4 (TLR4) expression, while facilitating the degradation of α-syn and its aggregates by promoting autophagy to protect against DA neuronal loss in vivo [141,142,143]. Moreover, microRNA-330 sponges could suppress LPS-induced polarization of microglia in a chronic neuroinflammatory model, prospectively by negatively modulating NF-κB activity and inflammatory cytokines such as iNOS by targeting SHIP1 in microglia [144]. MiR-let-7a expression was found to be decreased in a PD mouse model, and its overexpression negatively regulated α-Syn-induced BV-2 microglial cell activation and the release of pro-inflammatory factors by inhibiting STAT3 expression [145]. miR-155-5p reinforced the microglial activation and inflammatory factors release, enhanced oxidative stress and cell apoptosis, as well as alleviating the motivation of PD mice through targeting SOCS1 and Nrf2 [125]. Another anti-inflammatory component, Triptolide, was also verified to attenuate the inflammatory impairment via miR155-5p/SHIP1 pathway [146]. Interestingly, treatment of patients with PD with different medications (dopamine receptor agonists: pramipexole at a dosage of 1.5 mg/day or piribedil at a dosage of 150 mg/day, L-dopa at a dosage of 150–200 mg/day, and amantadine at a dosage of 300 mg/day) significantly increased the levels of miR-7, miR-9-5p, miR-9-3p, miR-129, and miR-132 [147]. Additionally, miR-19b may act as a potential biomarker for levodopa therapy by modulating ubiquitin-mediated proteolysis [148]. The miRNAs that have been shown to regulate downstream genes in PD inflammation are summarized in Table 3 and Figure 5.

### 4.4. ceRNA in the Pathogenesis and Neuroinflammation of PD

ceRNA has been widely associated with the pathogenesis of PD and neuroinflammation in PD. The lncRNA PLK2 has been found to facilitate a-syn aggregation-induced autophagy by sponging the miR-126/PLK2 signaling pathway [160]. Additionally, the lncRNA SNHG, considered a pivotal regulator in various cancers, is also associated with neurodegenerative diseases. SNHG1 promotes the aggregation of α-syn via the R-15b-5/SIAH1 axis [40]. Another signaling molecule, SNHG14, was verified to alleviate dopaminergic neuron injury and suppress α-syn aggregation via a sponge of miR-133b [161].

ceRNAs are crucial regulators of autophagy dysfunction in PD. As mentioned above, NEAT1 is attributable to autophagy in PD [162], and miR-34-5p/SYT1 has been demonstrated to serve as a sponge lncRNA NEAT1 and influence autophagy and apoptosis [163]. Moreover, the HOTAIR lncRNA could aggravate dopaminergic neuron autophagy by modulating the miR-221-3P/NPTX2, miR-874-5p/ATG10, and miR-126-5p/RAB3IP axis both in vitro and in vivo [164,165,166]. With miR-125-5p sponged, lncRNA BDNF-AS negative repression is likely to enhance cell proliferation as well as restrain autophagy and apoptosis both in vivo and in vitro [167].

Accumulating evidence indicates that ceRNAs are another inflammatory mediator network that activates inflammasomes during PD pathogenesis and induces microgliosis. Boros et al. and Simchovitz et al. detected the expression level of NEAT1, and the outcome showed an elevated level of NEAT1 in the peripheral blood cells and substantia nigra of patients with PD [168,169]. NEAT1 also exhibited a negative regulatory effect on neuron viability, activating the inflammatory response and oxidative stress by decreasing the expression of miR-1277-5P and consequently improving ARHGAP26 presentation [170]. The exhaustion of lncRNA NEAT1 reduced the release of TNF-α, IL-1β, and other neuroinflammatory cytokines and increased cell viability by targeting miR-124-3p and modulating PDE4B [171]. Wang et al. confirmed that miR-519a-3p/SP1 is another pathway regulated by NEAT1 [172]. It was discovered that NEAT1 was increasingly expressed in PD mouse and cell models, targeting the miR-212-3p/AXIN1, miR-212-5p/RAB3IP, miR-124/KLF4, and miR-374c-5p axis to suppress viability and promote the release of IL-1β, IL-6, and other inflammation factors [173,174,175,176], similarly participating in neuroinflammation, NEAT1 serves as a stimulative role of α-syn-induced activation of the NLRP3 inflammasome by targeting miR-1301-3p/GJB1 [177]. In addition to numerous animal and cell experiments, bioinformatics analysis also verified the crucial role of NEAT1 in ceRNAs in PD [144]. As a sponge for miR942-5, the lncRNA SRY-box regulates oxidative stress, inflammation, and neuronal apoptosis by indirectly mediating NAIF1 [178]. Additionally, lncRNA NORAD acts as a sponge of miR-204-5p by holding SLC5A3, and NORAD overexpression causes cytotoxicity, inflammatory response, and oxidative stress [179,180]. LncRNA AL049437 is a ceRNA for miR-205-5p, upregulated both in vivo and in vitro in PD models, and has been shown to induce neuroinflammation and oxidative stress in the disease by modulating MAPK1 [181]. Additionally, through the downregulation of miR-101, lncRNA Mirt2 inhibits the secretion of inflammatory cytokines and oxidative stress by blocking the TNF-α-triggered NF-κB/p38MAPK pathway [182]. With HOTTIP sponging to miR-615-3p, HOTTIP downregulation alleviates microglial activation, pro-inflammatory cytokine secretion and neuron death by lowering the expression of FOXO1 [183]. By binding to the miR-425-5P/TRAF5 axis, SNHG7 serves as a ceRNA to modulate inflammation and oxidative stress in cell and mouse models [184].

Abnormal apoptosis elicitation is involved in the pathogenesis of PD, and ceRNA networks have been linked to inappropriate apoptosis control, leading to dopamine neuron death. For instance, miR-51PA-3p can directly bind to NEAT, thereby inhibiting its expression, and its knockdown was verified to inhibit MPP(+)-induced repression of cell viability [172]. LINC00943 was found to be upregulated in MPP(+) cells, and increased the secretion of pro-inflammatory cytokines (TNF-α, IL-1β, and IL-6), oxidative injury, and apoptosis by inhibiting the miR-15b-5p/RAB3IP [185], miR-7b-5p/CXCL12 [186], and miR-142-5p/KPNA4/NF-κB axis [187]. The Wnt/β-catenin signal pathway plays a vital role in neurorestoration [188], while overexpression of lncRNA H19 protects dopaminergic neurons from loss by activating the Wnt/β-catenin pathway by repressing miR-301b-3p dependent HPRT1 expression [189]; moreover, H19 has been reported to attenuate neuronal injury by targeting the miR-585-3p signaling pathway [190]. Neural stem cells (NSCs) are now an effective therapeutic method for PD; miR-204-5p was found to increase in the course of NSCs differentiating into both neuron and astrocyte cells, as well as the repression of lncRNA ADNCR and TCF3, suggesting that lncRNA ADNCR may act as a sponge for miR-204-5p to regulate NSC differentiation by regulating TCF3 expression [191] (Table 4) (Figure 6).

### 4.5. Circulating miRNA in the Pathogenesis Neuroinflammation of PD

miRNAs are enriched in the brain tissue and play an essential role in the pathogenesis of Parkinson’s disease; therefore, they have been considered as potential diagnostic biomarkers. However, our capacity deficiency in accessing patients’ brain tissue from a specific region (such as the substantia nigra) is a major obstacle to the detection of miRNAs in PD. Therefore, the discovery of minimally invasive biomarkers is of great interest. Circulating cell-free miRNAs exist in the extracellular circulation, including plasma, serum, cerebrospinal fluid, saliva, urine, breast milk, and seminal plasma [205]. They are stable in severe environmental conditions and resistant to cellular RNase in circular systems owing to their existence in extracellular spaces [206,207]. Previous studies have proposed five different ways of miRNA transportation into extracellular fluid: bound with high-density lipoprotein complex particles in non-vesicle form, formed into complexes with multiple proteins (Ago2 and NPM1), packaged within exosomes, encapsulated within microvesicles (MVs), and accumulated in apoptotic bodies [208]. Exosomes have the potential to cross the blood–brain barrier via the membrane fusion process; therefore, miRNAs packaged within exosomes could also be considered as potential treatment targets [209,210,211,212]. Changing the expression profiles of miRNAs under pathogenetic and physiological states and their transportability through the brain–blood barrier have promoted the study of their potential application as biomarkers of Parkinson’s disease to diagnose at an early stage and identify new therapeutic targets (Table 5).

Emerging evidence has revealed that miR-34a-5p plays important roles in mammalian neurogenesis, synaptogenesis, and neural differentiation, and it has recently been regarded as a common dysregulated miRNA in different disorders of the CNS [213]. Grossi et al. reported that miR-34a-5p was significantly overexpressed in pure SEVs from the plasma of patients with PD compared to that in healthy controls. Additionally, pure SEVs miR-34a-5p levels were higher in patients with PD even at the initial stage of PD when the disease duration was less than 5 years. Its expression in pure SEVs revealed a good ability of this miRNA to distinguish patients with PD from control subjects, suggesting its potential use as a diagnostic marker at the molecular level [79].

Chen et al. developed a miRNA profiling strategy for circulating miRNAs isolated from the blood serum of 78 patients with PD and 78 normal controls, and seven of the most differentially expressed miRNAs were selected and further assessed by qRT-PCR. The results showed that elevated miR-133b and miR-221-3p levels differentiated early-stage PD from controls [214]. An interesting study demonstrated that the expressions of miR-106a-5p, miR-103a-3p, and miR-29a-3p in the experimental group were upregulated after physical exercise and were associated with cognitive improvement in patients with PD, suggesting evidence for the efficacy of physical exercise in maintaining or alleviating PD development [215]. Using a similar approach, Oliveira et al. identified and validated the low expression of serum miR-146a, miR-335-3p, and miR-335-5p in iPD (*n* = 45) and LRRK2-PD (*n* = 20) compared to the control group (*n* = 20) [216]. Doxakis integrated the studies and conducted bioinformative analysis, and the results showed 81 differentially expressed miRNAs from patients with PD and healthy controls. Gene function and pathway analysis of the deregulated mRNAs revealed biological pathways related to PD pathogenesis, including the FoxO signaling pathway, TNF signaling pathway, and ErbB signaling pathway, and indicated that autophagy participated in PD pathogenesis [217].

**Table 5 pharmaceuticals-15-00811-t005:** Circulating miRNA in the pathogenesis neuroinflammation of PD.

Sample Source	Species	miRNAs Status in PD	Method	Pilot Study	References
serum	human	↑/hsa-miR-7-5p, has-miR-22-3p, hsa-miR-136-3p, hsa-miR-139-5p, hsa-miR-330-5p, hsa-miR-433-3p, hsa-miR-495-3p	qRT-PCR	99 iPD vs. 101 HC	[218]
	human	↓/miR-96-5p	qRT-PCR	51 PD vs. 52 HC	[219]
	human	↓//hsa-miR-144-3p	NGS qRT-PCR	61 PD vs. 58 HC	[220]
	human	↓//hsa-miR-24-3p and hsa-miR-30c-5p	qRT-PCR	38 PD vs. 20 HC	[221]
	human	↑/miR-151a-5p, miR-24, mir-485-5p, mir-331-5p, and mir-214	qRT-PCR	209 PD vs. 60 HC	[222]
	human	↓//miR-214, miR-221, and miR-141	qRT-PCR	20 PD vs. 15 HC	[223]
	human	↓//miR-23b-3p	NGS qRT-PCR	22 PD vs. 9 HC	[224]
	human	↓//hsa-mir-4745-5p	qRT-PCR	12 PD vs. 12 HC	[225]
	human	↓//miR-374a-5p	qRT-PCR	68 PD vs. 50 HC	[226]
	human	↓//miR-21-3p, miR-22-3p and miR-223-5p	qRT-PCR	40 PD vs. 33 HC	[227]
	human	↑/miR-34a-5p	qRT-PCR	15 PD vs. 14 HC	[79]
	human	↑/miR-133b and miR-221-3p	qRT-PCR	151 PD vs. 138 HC	[214]
	human	↓//miR-124	qRT-PCR	25pPD vs. 21 HC	[228]
	human	↑/hsa-miR-374a-5p, hsa-miR-374b-5p	qRT-PCR	72 PD vs. 31 HC	[229]
	human	↑/miR-106a-5p, miR-103a-3p, miR-29a-3p	qRT-PCR	8PD (after exercise) vs. 8PD (before exercise)	[215]
	human	↓//miR-132-3p, miR-146-5p	qRT-PCR	82 PD vs. 44 HC	[73]
	human	↑/miR-27a-3p, miR-584-5p	NGS	7 PD vs. 24 HC	[230]
	human	↓//miR-146a, miR-335-3p, miR-335-5p	qRT-PCR	20 iPD vs. 20 HC	[220]
	human	↓//miR-150	qRT-PCR	80 PD vs. 80 HC	[134]
	human	↑/miR-330-5p, miR-433-3p, miR-495-3p	qRT-PCR	108 PD vs. 92 HC	[231]
	human	↓//miR-29	qRT-PCR	37 PD vs. 40 HC	[74]
	human	↓//miR-218, miR-124, miR-144	qRT-PCR	15 PD vs. 10 HC	[126]
	human	↑/miR-132	qRT-PCR	269 PD vs. 222 HC	[75]
	human	↑/miR-105-5p	qRT-PCR	319 PD vs. 273 HC	[77]
CSF	human	↑/miR-151a-5p, miR-24, mir-485-5p, mir-331-5p, and mir-214	qRT-PCR	209 PD vs. 60 HC	[222]
	human	↑/miR-7-5p, miR-331-5p, miR-145-5p	qRT-PCR	10 PD vs. 10 HC	[103]
	human	↓//hsa-miR-626	qRT-PCR	15 PD vs. 16 HC 20 PD vs. 27 PD HC	[100,232]
saliva	human	↓//miR-29a-3p, miR-29c-3p	qRT-PCR	5 PD vs. 5 HC	[233]
	human	↓//miR-153, miR-223	qRT-PCR	83 PD vs. 77 HC	[72]

## 5. circRNA

### 5.1. Introduction

circRNA was first discovered in viroids by Sanger in 1976 [234]. It is a type of ncRNA molecule with a closed-loop structure. There is no 5′-cap arrangement and a 3′-poly tail; therefore, it is not easily degraded by RNase [235]. It primarily exists in the cytoplasm and in exosomes [236]. circRNA is formed by reverse splicing of pre-mRNA, and its circularization process is regulated by RNA polymerase II (Pol II), cis-acting elements, trans-acting factors, and RNA-binding proteins [235]. According to their formation mechanism, circRNAs can be divided into three categories: single or multiple exons (circRNA), formed by the failure of debranching of the intron lasso (circRNAs), and both introns and exons (EIciRNAs) [237]. The degradation mechanism of circRNAs is still poorly understood, and it has been suggested that they may be degraded by small RNAs [238].

### 5.2. circRNA in the Pathogenesis of PD

As previously mentioned, mitochondrial dysfunction, oxidative stress, altered protein handling, and neuroinflammation are involved in the pathogenesis of PD. Some studies have shown that circRNAs participate in at least one of these pathological processes. Kong et al. used RNA sequencing and found that hsa_circ_0001451, hsa_circ_0001772, and hsa_circ_0036353 were downregulated in the peripheral blood RNAs of patients with PD, encoded for FBXW7, RBM33, and SIN3A [239]. Ravanidis et al. performed qRT-PCR of total RNA in patients with PD and healthy controls and found that Hsa_circ_0000497 was downregulated in the peripheral blood of patients with PD, which is encoded by the SLAIN motif family member-1 (SLAIN1) gene, microtubule-associated proteins, and is essential for axon elongation in neuronal development. Therefore, the deregulation of hsa_circ_0000497 deregulates offline cytoskeletal dynamics, which is expected to diminish intracellular signaling pathways and has been regarded as a critical factor in the pathogenesis of multiple neurodegenerative diseases, including PD [240].

### 5.3. circRNA in Neuroinflammation of PD

The relationship between circRNAs and inflammation is now gaining recognition, and some studies have revealed that circRNAs participate in diabetes mellitus [241], atherosclerosis [242], osteoarthritis [243], and spinal cord injury [244], all of which are associated with immunoinflammatory effects. However, circRNA profiling and the regulatory mechanisms of neuroinflammation in PD remain unclear.

### 5.4. ceRNA in the Pathogenesis and Neuroinflammation of PD

circRNAs can relieve the inhibitory effect of miRNA on mRNA because they are rich in miRNA binding sites. In addition, circRNA does not contain 3′ and 5′ free ends, so it is not easily degraded by exonucleases, enhancing its function as a miRNA sponge [245]. circRNAs are known to act as miRNA sponges, circSNCA was confirmed to be another miR-7 repressor, and it was found to be downregulated in a pramipexole-worked PD cell model as well as an inhibitor of α-syn deposition, while miR-7 was upregulated [199]. Besides being associated with α-syn aggregation, circRNAs are also attributable to autophagy in PD. CircDLGAP4 acts as a sponge for miR-134-5p and plays a role in suppressing its functional activity of miR-134-5p. circDLGAP4 was reported to be downregulated in an MPTP-induced PD animal model, which could promote autophagy, inhibit mitochondrial dysfunction, and suppress apoptosis by modulating the activation of CREB by regulating miR-134-5p [200]. Similarly, miR-29c-3p, which targets and inhibits the expression of circSAMD4A, is involved in apoptosis and autophagy of dopaminergic neurons by regulating the AMPK/mTOR cascade [201]. Additionally, in the worm PD model, cirzip-2 was found to target miR-60-3p, which contains M60.4, igeg-2, got-2.2, K07H8.2, asns-2, pkg-2, ZK470.2, and idhg-1 genes, which were found to be downregulated in the PD model [202]. Additionally, circSLC8A1 was verified to correlate with oxidative stress in Parkinsonism via binding to miR-128, regulating the microRNA effector protein Ago2 in the PD cell model [203]. Simultaneously, circ_0070441 can directly bind to miR-626, thereby indirectly regulating IRS2 expression, and its upregulation was verified to aggregate MPP(+)-induced neurotoxicity in a PD cell model [204]. However, there is still a controversial issue about whether circRNAs can serve as miRNA sponges because some studies have revealed that most circRNAs do not have the function of sponging miRNAs [210] (Figure 6).

## 6. ncRNAs in Treatment of Parkinson’s Disease

ncRNA therapeutics comprise various groups of oligonucleotide-based drugs such as antisense oligonucleotides, small interfering RNAs (siRNAs), short hairpin RNAs (shRNAs) and clustered regularly interspaced short palindromic repeats (CRISPR/Cas) that can be designed to selectively interact with drug targets. However, as a result of the blood–brain barrier hindering medication delivery to the CNS, the drug delivery system is key to the envisioned drug role for ncRNAs. The drug delivery system consists of exogenous vectors (lentiviral vectors, adenoviral vectors, nanomedicines, liposomes, and lipid NPs) and endogenous vectors (exosomes). Owing to the disadvantages of toxicity and immunogenicity associated with viral vectors, major attention has been focused on the development of non-viral vectors, such as polymeric NPs, which may result in a safer and less toxic gene delivery method [245,246,247,248,249,250,251].

As drug carriers, exosomes have the advantages of optimized biocompatibility, blood–brain barrier penetrability, metabolic stability, and target specificity. Exosomes have been successfully loaded with catalase, dopamine, and small interfering RNA for PD treatment and have shown significant therapeutic effects. In 2014, Cooper et al. found that siRNA-loaded exosomes could significantly decrease the expression levels of α-syn mRNA and protein compared with that seen in healthy controls [251]. In 2015, Haney et al. delivered exosomes via nasal injection into a PD mouse model and demonstrated specific neuroprotective effects. Recently, Kojima et al. reported that catalase mRNA-loaded exosomes effectively attenuated neurotoxicity and neuroinflammation in PD mice [252,253]. Therefore, it is worthwhile to develop a drug delivery system for exosomes for the treatment of PD.

Nanomedicine has also been studied for controlled delivery of genes to the brain. One particularly interesting example of the development of nanomedicines for microRNA delivery involves the use of polymeric nanoparticles for miR-124. It has been demonstrated that stereotactic injection into the right lateral ventricle of miR-124-loaded NPs attenuates neurogenesis in the subventricular zone, promoting migration and integration of mature neurons into the lesioned striatum of 6-OHDA–treated mice and improving motor symptoms [254].

## 7. Conclusions

In conclusion, this manuscript identifies and describes the multiform ncRNAs, including miRNAs, lncRNAs, and circRNAs, and their regulatory mechanisms participating in the pathogenesis of PD. Increasing evidence indicates that ncRNAs play a pivotal role in neuroinflammation and neurodegeneration in the CNS. However, more extensive ncRNA profiling data and experiments are required to ultimately invert transcriptional analysis into clinical treatment and diagnosis. Moreover, further research is needed to fully understand the intricate and unsuspected correlation between ncRNAs, proteins, and cellular physiology and pathology in PD. Recent studies have paid more attention to miRNAs and lncRNAs; the understanding of circRNAs is supposed to be deeply studied, especially neuroinflammation in PD, given the new technologies, such as second-generation sequencing. At the same time, it is essential to further clarify the spatial and temporal distribution of ncRNAs in PD. Silico and systematic analyzes must be performed to recognize the complicated network between ncRNAs and proteins in the pathogenesis of PD. Since pharmacotherapy, such as L-dopa and peripheral dopa decarboxylase inhibitors, can relieve motor symptoms but cannot restrain the development of PD, ncRNA-related treatment may become a potential and novel therapy in the development of PD on account of this evidence, such as cell replacement therapy and molecular targeting treatment. The ncRNA miR-204-5p increases during the differentiation of NSCs into both neurons and astrocytes. Another study showed that astrocytes can be induced to form dopamine neurons via NEUROD1, ASCL1, and LMX1A [143]. In sum, identifying ncRNAs involved in PD progression may lead to more effective diagnosis and treatment.

## Figures and Tables

**Figure 1 pharmaceuticals-15-00811-f001:**
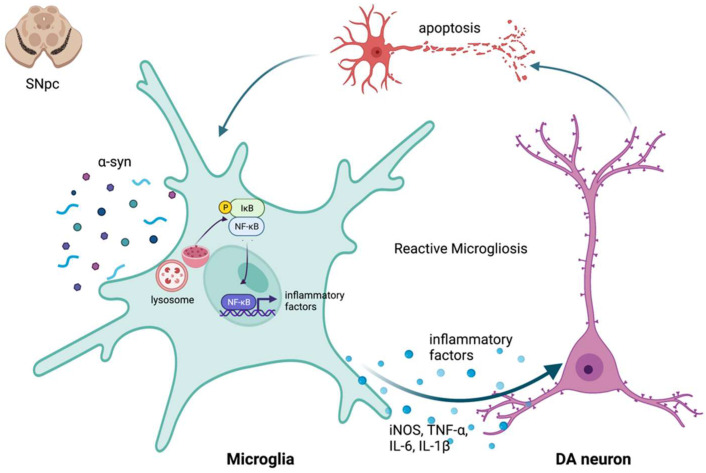
The self-propelled deterioration cycle in PD. Microglia are active under the pathogenic conditions of PD and release anti-inflammatory cytokines to heal the tissues, saving neurons from apoptosis or death. Continuous stimulation of pathogenic factors, on the other hand, increases the number of toxic phenotypes of microglia, resulting in the release of a significant number of inflammatory cytokines such TNF-, IL-1, iNOS, and IL-6, all of which lead to neuronal damage. Furthermore, damaged or dead DA neurons can directly activate microglia, resulting in an increase in reactive oxygen species (ROS) and pro-inflammatory cytokines. Microglia activation and DA neuronal injury thus form a self-propelled degeneration cycle in PD.

**Figure 2 pharmaceuticals-15-00811-f002:**
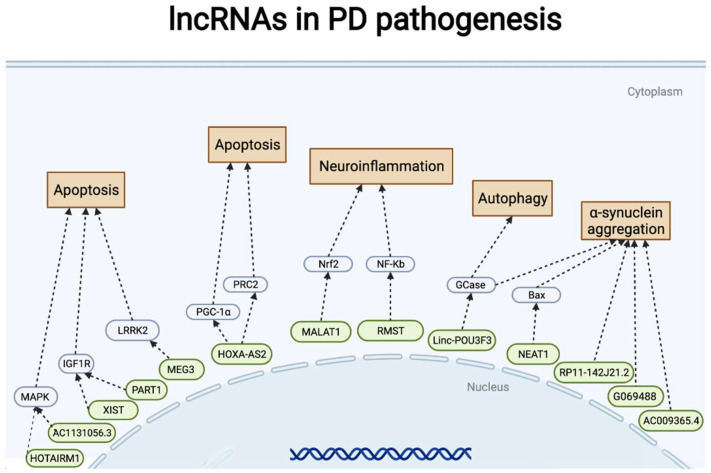
LncRNA representative signal pathways involved in Parkinson’s disease pathogenesis and neuroinflammation. Straight arrows indicate lncRNAs regulate gene (protein) expression in PD pathogenesis. Two consecutive arrows mean that there may be other participants in the process.

**Figure 3 pharmaceuticals-15-00811-f003:**
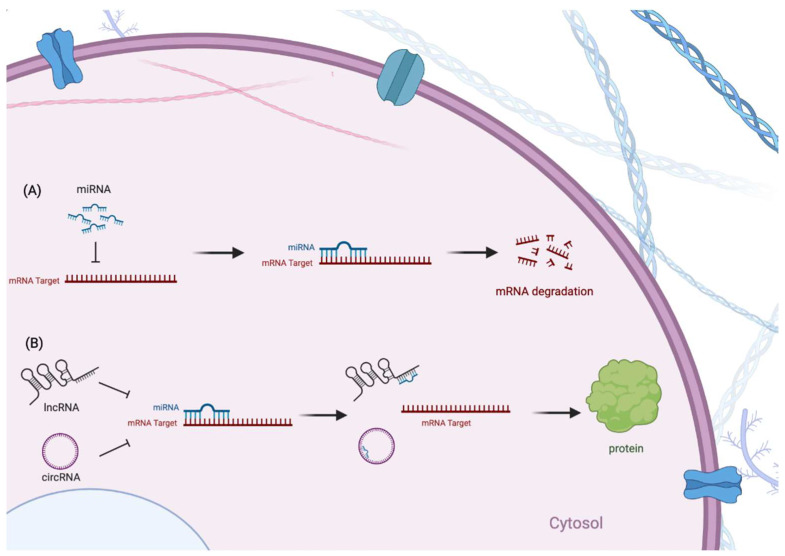
The targeted regulation mechanism of miRNA and competitive endogenous RNAs (ceRNAs). (**A**) The regulation function of miRNAs is to target mRNA molecules with complementary sequences. As a result, the genes can be silenced by cleavage of the mRNA strand. (**B**) LncRNA and circRNA combine with miRNA so that mRNAs are protected from degradation and consequently translate to proteins.

**Figure 4 pharmaceuticals-15-00811-f004:**
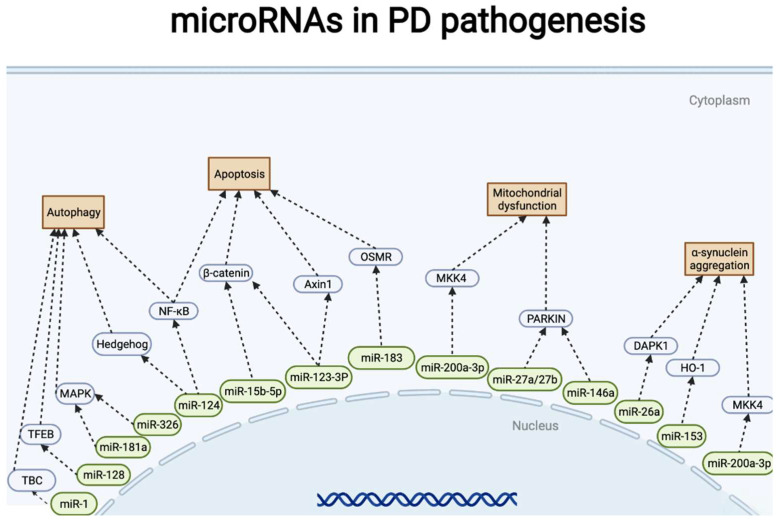
miRNA representative signal pathways involved in Parkinson’s disease pathogenesis. Straight arrows indicate miRNAs regulate gene (protein) expression in PD pathogenesis. Two consecutive arrows mean that there may be other participants in the process.

**Figure 5 pharmaceuticals-15-00811-f005:**
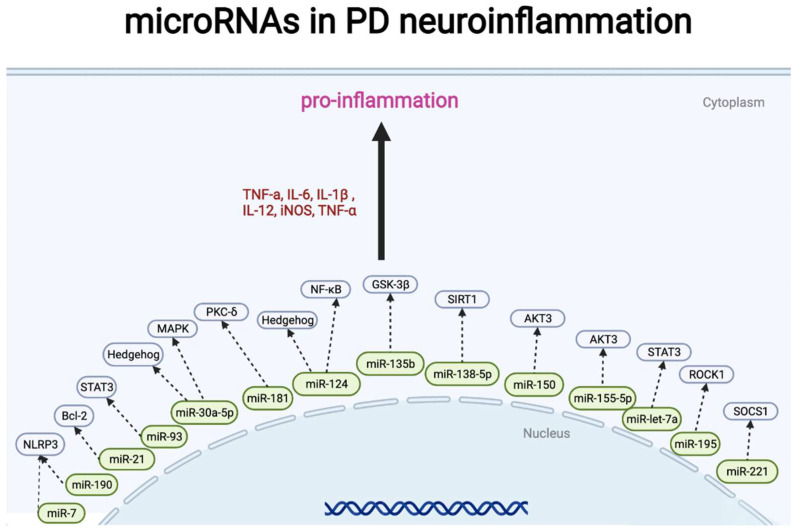
miRNA representative signal pathways involved in Parkinson’s disease neuroinflammation. Straight arrows indicate miRNAs regulate gene (protein) expression in PD neuroinflammation. Two consecutive arrows mean that there may be other participants in the process.

**Figure 6 pharmaceuticals-15-00811-f006:**
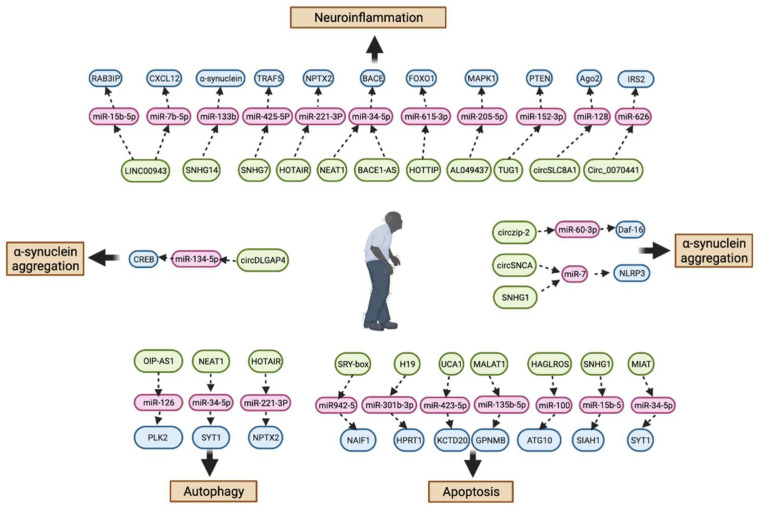
ceRNA representative signal pathways involved in Parkinson’s disease pathogenesis. Straight arrows indicate ceRNAs regulate gene (protein) expression in PD pathogenesis. Two consecutive arrows mean that there may be other participants in the process.

**Table 1 pharmaceuticals-15-00811-t001:** lncRNA in pathogenesis and neuroinflammation of PD.

Species of lncRNAs	Changes in lncRNAs Levels in PD’s Brain, CSF and Serum	Changes in lncRNAs Levels in Genetic Mouse Models and Cell Models for PD	Response of lncRNAs to PD “Triggers” In Vitro (Exposure Time if Relevant)	lncRNAs Target Genes (Experimentally Validated)
H19	↓/PD’s brain [26]			
lincRNA-p21	↑/PD’s brain [26]			
LINC-PINT	↑/PD’s brain [26]			
Malat1	↑/PD’s brain [26]			
SNHG1	↑/PD’s brain [26]			
HOTAIRM1	↑/PD’s circulating leukocytes [23]		↑/apoptosis ↑/neuroinflammation	MAPK, Jak-STAT
AC1131056.3	↑/PD’s circulating leukocytes [23]		↑/apoptosis ↑/neuroinflammation	MAPK, Jak-STAT
XIST	↓/PD’s serum [27]		↓/apoptosis	hsa-miR-133b/IGF1R
PART1	↓/PD’s serum [27]		↓/apoptosis	hsa-miR-133b/IGF1R
rs13388259	↓/PD’s serum [28]		↑/apoptosis	HNF4A
NEAT1		↑/MPTP induced C57BL/6 mice [29]	↑/α-synuclein aggregation ↑/apoptosis	Bax/Bcl-2 caspase-3
G069488			↑/α-synuclein aggregation	
RP11-142J21.2			↑/α-synuclein aggregation [21]	
AC009365.4.			↑/α-synuclein aggregation [21]	
Linc-POU3F3	↑/L1CAM exosome in PD plasma [30]		↑/autophagy ↑/α-syn concentrations	GCase
MEG3	↓/PD’s serum [35]	↓/MPP+ treated SH-SY5Y cells [32,33]	↓/apoptosis	LRRK2
HOXA-AS2	↑/PD‘s PBMCs [41]		↑/neuroinflammation ↑/microglial activation	↓/PGC-1α ↑/PRC2
MALAT1		↑/MPTP-induced PD mice ↑/LPS/ATP-induced microglia cells [42]	↑/inflammasome activation ↑/reactive oxygen species (ROS)	↓/Nrf2
RMST		↑/brain SN of PD rats	↑/oxidative stress ↑/apoptosis	↑/TLR/NF-Κb
Lnc-MKRN2-42:1	↑/PD’s serum [36]			
LINC-PINT	↑/PD’s serum [37]		↓/cellular survival ↑/oxidative stress	
UCA1		↑/PD mice brain ↑/MPP+-induced SH-SY5Y cells [39,40]	↑/caspase-3 activation ↑/apoptosis	↑/SNCA
HOTAIR		↑/PD mice brain ↑/MPP+-induced SH-SY5Y cells [41]	↑/caspase-3 activation ↑/apoptosis	↑/LRRK2
GAS5		↑/old mouse brain [38]	↓/cell cycle progression ↑/apoptosis	

**Table 2 pharmaceuticals-15-00811-t002:** microRNAs in PD pathogenesis.

Species of miRNAs	Changes in miRNA Levels in PD’s Brain, CSF and Serum	Changes in miRNA Levels in Genetic Mouse Models and Cell Models for PD	Response of miRNA to PD “Triggers” In Vitro (Exposure Time if Relevant)	miRNA Target Genes (Experimentally Validated)
miR-1	↓/CSF		↓/autophagy	↑/TBC-7 ↑/TBC1D15 [92,93]
miR-1-3p				FAIM [105]
miR-15b-5p		↓/SH-SY5Y cells	↑/apoptosis	↑/Akt3 ↑/CSK-33β/β-catenin [97]
miR-26a	↓/PBMCs	↓/C57 BL/6	↑/α-syn	↑/DAPK1 [106]
miR-27a/27b	↑/midbrain		↑/mitochondrial fragmentation ↑/ROS	↓/PARKIN ↓/PINK1 [6]
miR-29c	↑/serum [78]			
miR-29c-3p		↓/DA neuron	↓/autophagy	↑/TET2
miR-29b	↓/serum [74,82]			
miR-34a-5p	↑/EVs [79]	↑/SH-SY5Y cells	↑/ER stress	↓/IRE1α [107]
miR-103		↓/MN9D cells ↓/C57BL/6 mice	↑/LC3-II ↑/p62	↑/CDK5R1/CDK5 [108]
miR-105-5p	↑/serum [77]			
miR-107		↓/MN9D cells ↓/C57BL/6 mice	↑/LC3-II ↑/p62	↑/CDK5R1/CDK5 [108]
miR-123-3P	↓/hippocampal tissue	↓	↑/apoptosis	↑/Axin1 ↓/Wnt/β-catenin [109]
miR-124	↓/brain	↓/C57BL/6 mice	↑/cytokines ↑/apoptosis ↓/autophagy	↑/MEKK3/NF-κB [8] ↑/62/p38 [110] ↑/Hedgehog [111]
miR-125b-5p				
miR-128	↑/brain	↑/DA neuron	↓/autophagy ↑/α-syn	↓/TFEB [90]
miR-132-3p	↓, ↑/serum [73,74,75]			
miR-133a		↓/PC-12 cells	↑/apoptosis	↑/RAC1 [112]
miR-133b				FAIM [105]
miR-137	↑/plasma		↑/ROS	↓/OXR1 [113]
miR-138-5p		↑/SH-SY5Y cells	↑/TNF-α ↑/IL-1β ↑/ROS	↓/SIRT1 [114]
miR-144	↓/brain		↓/autophagy ↑/mitochondrial fragmentation	↑/mTOR [115] ↑/β-amyloid precursor protein [89]
miR-146a		↑/SH-SY5Y cells	↑/mitochondrial fragmentation ↑/ROS	↓/PARKIN [87]
miR-146-5p	↓/serum [73,74]			
miR-153	↓/brain ↓/serum [72,84] ↓/saliva		↑/α-syn	↑/HO-1 [84]
miR-155		↑/C57BL/6 mice	↑/α-syn ↑/iNOS	↑/MHCII [80]
miR-155-5p	↑/PBMCs [107]		↑/α-syn ↑/iNOS [81]	
miR-181a	↓/serum	↓/SK-N-SH	↑/apoptosis ↓/autophagy	↑/p38MAPK/JNK [116]
miR-183	↑/brain	↑/substantia nigra neurons	↑/apoptosis	↓/OSMR [117]
miR-185		↓	↑/ROS	↓/PI3K/AKT ↓/IGF1 [98]
miR-195	↑/serum		↑/neuroinflammation	↑/ROCK1 [78,118]
miR-200a-3p		↓/MPP-ADEXx	↑/apoptosis	↑/MKK4 [119]
miR-204-5P	↑/brain		↑/α-syn [104]	↑/DYRK1A [85]
miR-206				FAIM [105]
miR-216			↓/apoptosis	↓/Bax [120]
miR-217		↑/SH-SY5Y	↑/TNF-α ↑/IL-1β ↑/ROS	↓/SIRT1 [84]
miR-218-5p	↓/brain	↓/brain SN of PD rats	↑/apoptosis ↑/ROS ↑/NF-Κb	↑/LASP1 [121]
miR-223	↓/brain [72] ↓/serum↓/saliva			
miR-291			↓/ROCK2 [122]	
miR-326	↓/brain		↓/autophagy ↑/iNOS ↑/apoptosis ↑/α-syn	↑/XBP1 [94,95] ↑/MAPK/KLK7
miR-331-5p	↓/CSF [103]			
miR-342-3p		↑/C57BL/6 mice	↑/apoptosis	↓/PAK1 ↓/Wnt [123]
miR-380-3p		↑/N2a	↑/apoptosis	↓/Sp3 [96]
miR-486-5p	↑/colonic biopsies [76]			
miR-599	↓/brain			↑/LERK2 [80]
hsa-miR-626	↓/CSF [100]			
hsa-miR-19b-3p	↑/CSF [124]			

**Table 3 pharmaceuticals-15-00811-t003:** microRNAs in PD neuroinflammation.

Species of miRNAs	Changes in miRNA Levels in PD’s Brain, CSF and Serum	Changes in miRNA Levels in Genetic Mouse Models and Cell Models for PD	Response of miRNA to PD “Triggers” In Vitro (Exposure Time if Relevant)	miRNA Target Genes (Experimentally Validated)
miR-7	↓/PD’s serum [128]	↓/MPTP/p-treated mice↓/A53T tg/tg mice [128]	↓/IL-1β↓/α-syn aggregation [142] ↑/autophagy [142]	↓/NLRP3 [148] ↓/TLR4 [141]
miR-let-7a		↓/C57BL/6 mice ↓/LPS-exposed BV2 cells	↓/microglia activation ↓/TNF-a, IL-6, IL-1b, and IL-12	↓/STAT3 [145]
miR-21		↑/MPP(+) treated MES23.5 cells [137]	↑/iNOS ↑/IL-1β, IL-6 and TNF-α ↑/apoptosis	↓/Bcl-2
miR-29a-3p			↑/microglial motility ↑/phagocytosis	↑/ROCK/CDC42 [129]
miR-30a-5p		↓/microglial [149]	↓/TNF-α, IL-1β and IL-10	↓/Neurod 1 ↓/MAPK/ERK
miR-93		↓/LPS-exposed BV2 cells [150]	↓/microglial activation ↓/iNOS, IL-6 and TNF-α	↓/STAT3
miR-99a		↑/C57BL6 mice microglial [125]		
miR-103a-3p			↑/microglial motility ↑/phagocytosis	↑/ROCK/CDC42 [129]
miR-181		↓/BV2 cells [34]	↓/iNOS, NO and ROS ↓/microglial motility	↑/PKC-δ
miR-124	↓/PD’s brain [8]	↓/BV2 cells [8]	↓/neuroinflammation [111]↓/neuronal death [111]↓/apoptosis [8] ↓/TNF-α, iNOS, and IL-1b [110] ↑,↓/autophagy [110]	↓/MEKK3/NF-κB [8] ↓/p62/p-38 [129] ↓/Hedgehog [130]
miR-125b-5p		↑/C57BL6 mice microglial [125]		
miR-128		↑/A9,A10 DA neurons [90,151]	↑/α-syn aggregation	↓/p38 ↓/TFEB
miR-129-5p		↓/C57BL6 mice [152]	↓/inflammation↓/blood–spinal cord barrier (BCSB)	↓/HMGB1 ↓/TLR3
miR-135b		↓/SH-SY5Y cells [128]	↑/TNF-α, IL-1β ↑/apoptosis ↓/pyroptosis [153]	↑/GSK-3β ↓/FoxO1 [153] ↓/TXNIP, NLRP3, Caspase-1 [153]
miR-138-5p		↑/MPP(+) induced SH-SY5Y cells [134]	↑/iNOS ↑/IL-1β and TNF-α	↓/SIRT1
miR-144	↓/PD’s brain			↑/NF-κB [126]
miR-146a	↑/PD’s PBMCs [81,140]		↑/microglial activation	
miR-150	↓/PD‘s serum	↓/BV2 cells [154]	↓/IL-1β, IL-6 and TNF-α [154,155]	↓/AKT3
miR-155	↑/PD’s PBMCs [81]	↑/C57BL6 mice [82]	↑/iNOS [82] ↑/microglial activation	↑/MHCII ↓/SOCS1
miR-155-5p	↑/PD’s PBMCs [81]	↑/C57BL6 mice [124]	↑/microglial activation↑/oxidative stress ↑/apoptosis ↑/TNF-α and IL-1β	↓/SOCS1 ↓/Nrf2
miR-188-3p	↓/PD’s serum		↓/pyroptosis [156] ↓/autophagy	↓/CDK50 ↓/NLRP3
miR-190		↓/LPS-induced BV2 cells [157]	↓/iNOS, IL-6 and TGF-β ↑/IL-10	↓/NLRP3
miR-195		↓/LPS-induced BV2 cells [118]	↓/IL-6 and TNF-α ↑/IL-4 and IL-10	↓/ROCK1
miR-217		↑/MPP(+) induced SH-SY5Y cells [134]	↑/iNOS ↑/IL-1β and TNF-α	↓/SIRT1
miR-218	↓/PD’s brain			↑/NF-κB [147]
miR-221		↑/CCI-induced rat model [158] ↓/6-OHDA treated PC12 pheochromocytoma cells [159]	↑/TNF-α,IL-1β, and IL-6 ↓/apoptosis	↓/SOCS1 ↑/NF-κB ↑/p38 MAPK ↓/PTEN
miR-330		↓/LPS-induced BV2 cells [144]	↓/microglial polarization ↓/iNOS	↓/NF-κB ↑/SHIP1 and Arg1
miR-342-3p		↑/C57BL6 mice microglial [131]		
miR-375		↓/6-OHDA treated Wistar rats substantia nigra [136]	↓/TNF-α, IL-6 and IL-1β ↓/SOD and GSH-Px ↑/MDA	↓/SP1
miR-3473b		↑/LPS treated BV2 cells [138] ↑/C57/BL6 mice	↑/microglial motility ↑/autophagy	↓/TREM2/ULK1
miR-7116-5p		↓/C57BL6 mice microglia [127]	↓/IL-1β, IL-6,TNF-α and iNOS	

**Table 4 pharmaceuticals-15-00811-t004:** ceRNA in pathogenesis and neuroinflammation of PD.

Non-Coding RNA	Species of ncRNAs	Expression	Related Genes	Functional Role in PD	References
lncRNA	OIP-AS1	↓	miR-126/PLK2	↓/α-synuclein aggregation ↑/autophagy	[160]
	SNHG1	↑	miR-15b-5/SIAH1 miR-7/NLRP3 miR-181a-5p/CXCL12 miR-125B-5p/MAPK1 miR-216A-3P/Bcl-2 miR-221/222/p27/mTOR miR-15b-5p/GSK3β miR-153-3p/PTEN/AKT/mTOR	↑/α-synuclein aggregation ↑/apoptosis ↑/microglial activation	[40,44,45,46,47,48]
	SNHG14	↑	miR-133b/α-synuclein miR-124-3p/KLF4	↑/DA neuron injury ↑/α-synuclein aggregation ↑/apoptosis ↑/inflammation	[161]
	SNHG7	↑	miR-425-5P/TRAF5	↑/neuronal apoptosis ↑/TH-positive cell loss ↑/microglial activation ↑/oxidative stress ↑/neuroinflammation	[184]
	NEAT1	↑	miR-34-5p/SYT1 miR-51PA-3p	↑/autophagy ↑/apoptosis ↓/cell proliferation	[162,163,172]
	HOTAIR	↑	miR-221-3P/NPTX2 miR-874-5p/ATG10 miR-126-5p/RAB3IP miR-326/ELAVL1	↑/autophagy ↑/apoptosis ↓/cell proliferation ↑/NLRP3 inflammasome activation	[164] [165] [166] [192]
	BDNF-AS	↑	miR-125b-5p	↑/autophagy ↑/apoptosis ↓/cell proliferation	[167]
	NEAT1	↑	miR-1277-5P/ARHGAP26 miR-124-3p/PDE4B miR-519a-3p/SP1 miR-212-3p/AXIN1 miR-212-5p/RAB3IP miR-124/KLF4 miR-374c-5p miR-1301-3p/GJB1	↑/neuroinflammation (IL-6,TNF-α,IL-1β) ↓/neuron viability ↑/apoptosis ↑/NLRP3 inflammasome activation	[163,170,171,172,173,174,176]
	SRY-box	↑	miR942-5/NAIF1	↑/apoptosis ↑/cleaved caspase-3 protein expression ↑/TNF-α, IL-1β, ROS and LDH	[178]
	NORAD	↓	miR-204-5p/SLC5A3	↓/cytotoxicity ↓/inflammatory response ↓/oxidate stress (ROS)	[179,180]
	AL049437	↑	miR-205-5p/MAPK1	↑/cytotoxicity↑/inflammatory response ↑/oxidate stress (ROS)	[181]
	Mirt2	↓	miR-101/NF-κB/p38MAPK	↓/inflammatory response (IL-6,TNF-α,IL-1β) ↓/oxidate stress (ROS) ↓/apoptosis	[182]
	HOTTIP	↑	miR-615-3p/FOXO1	↑/microglial activation, ↑/proinflammatory cytokine secretion (IL-lβ, IL-6, IL-18, TNF-α, iNOS, COX2, NF-κB) ↑/apoptosis	[183]
	LINC00943	↑	miR-15b-5p/RAB3IP miR-7b-5p/CXCL12 miR-142-5p/KPNA4/NF-κB	↑/TNF-α, IL-1β and IL-6 ↑/oxidative injury ↑/apoptosis	[185,186,187]
	BACE1-AS	↑	miR-34b-5p/BACE	↑/TNF-α, IL-1β and IL-6 ↑/oxidative injury ↑/apoptosis	[193]
	H19	↓	miR-301b-3p/HPRT1/Wnt/β-catenin miR-585-3p	↓/neuron loss ↓/neuronal injury	[189,190]
	UCA1	↑	miR-423-5p/KCTD20	↑/cytotoxicity ↑/apoptosis	[194]
	SOX21-AS1	↑	miR-7-5p/IRS2	↑/cell injury	[195]
	MALAT1	↑	miR-135b-5p/GPNMB	↓/cell proliferation ↑/apoptosis	[196]
	TUG1	↑	miR-152-3p/PTEN	↑/cell apoptosis ↑/oxidative stress ↑/neuroinflammation	[197]
	HAGLROS	↑	miR-100/ATG10 PI3K/Akt/Mtor	↑/cell apoptosis ↓/proliferation	[154]
	MIAT	↓	miR-34-5p/SYT1	↓/apoptosis	[198]
	HOXA11-AS	↑	miR-124-3p/FSTL1	↑/NLRP3 inflammasome activity ↑/microglial activation	[50]
	Linc-p21	↑	miR-625/TRPM2 miR-181/PKC-δ	↑/oxidative stress	[26,34]
	ADNCR		miR-204-5p/TCF3		
	T199678	↓	miR-101-3p	↓/oxidative stress ↓/apoptosis	[31]
	GAS5	↑	miR-223-3p/NLRP3	↑/microglial activation, ↑/apoptosis	[35]
circRNA	circSNCA	↑	miR-7	↑/cell apoptosis ↑/α-synuclein aggregation ↓/autophagy	[199]
	circDLGAP4	↓	miR-134-5p/CREB	↑/autophagy ↓/mitochondrial dysfunction ↓/apoptosis	[200]
	circSAMD4A	↑	miR-29c-3p/AMPK/mTOR	↑/autophagy ↑/apoptosis	[201]
	circzip-2	↓	miR-60-3p/Daf-16	↓/α-synuclein expression ↓/ROS	[202]
	circSLC8A1	↑	miR-128/Ago2	↑/oxidative stress	[203]
	Circ_0070441	↑	miR-626/IRS2	↑/cell apoptosis ↑/inflammation	[204]

## Data Availability

Data sharing not applicable.

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
