# Peer review of "The Role of Non-Coding RNAs in the Pathogenesis of Parkinson’s Disease: Recent Advancement"

_pharmaceuticals, 2022, doi:10.3390/ph15070811_

Round 1

Reviewer 1 Report

There are a large number scientifically unclear and incorrect statements in the paper, many due to badly written English.  I started to list them, but frankly, I became overwhelmed, there were too many. The subject of non-coding RNAs in the pathogenesis of Parkinson's disease is important and worthy of review, and the authors' reference list and tables are quite impressive.  I hope they get the help they need to fix the English problems.

Author Response

Dear reviewer,

Firstly, we would like to thank you for your kind letter and for reviewers’ constructive comments concerning our article (Manuscript No.: pharmaceuticals-1726402). These comments are all valuable and helpful for improving our article. All the authors have seriously discussed about all these comments. According to reviewers’ comments, we have tried our best to modify our manuscript to meet with the requirements of your journal. In this revised version, changes to our manuscript within the document were all highlighted by using yellow colored text. For example, XXXX means newly added content. Point-by-point responses to the reviewers are listed below this letter

General:

There are a large number scientifically unclear and incorrect statements in the paper, many due to badly written English. I started to list them, but frankly, I became overwhelmed, there were too many. The subject of non-coding RNAs in the pathogenesis of Parkinson's disease is important and worthy of review, and the authors' reference list and tables are quite impressive. I hope they get the help they need to fix the English problems.

Response: Thank you very much for your question and careful look. According to your suggestion, we sent our manuscript to Elseveir for language editing in order to improve written English.

Reviewer 2 Report

The review by Zhang et al. provides an overview of the role of ncRNAs in PD pathogenesis.

1.     The readability of the discussion should be improved by depicting n the diagram the complexity of the links between the different types of ncRNAs and molecular pathways–targets involved in PD.

2.     As ncRNAs can serve as potential non-invasive biomarkers of PD, circulating brain‑derived ncRNAs should be discussed in the context of specificity and BBB permeability.

3.     Delivery options may be discussed as potential targeted treatment in the early stage of PD has been mentioned.

Addressing the above issues will make this paper more impactful.

Round 2

Reviewer 1 Report

This review of non-coding RNA in Parkinson's disease is very comprehensive and the work to improve the English is appreciated.  Unfortunately, there are still a large number of inaccurate and unclear statements in section 2, Pathogenesis of PD.  This is section is not necessary for the review since is does not directly pertain to non-coding RNA. In my opinion, if section 2 is removed, the manuscript would be acceptable for publication.

Author Response

Dear reviewers

Firstly, we would like to thank you for your kind letter and for reviewers’ constructive comments concerning our article (Manuscript No.: pharmaceuticals-1726402). These comments are all valuable and helpful for improving our article. All the authors have seriously discussed about all these comments. According to reviewers’ comments, we have tried our best to modify our manuscript to meet with the requirements of your journal.

General:

This review of non-coding RNA in Parkinson's disease is very comprehensive and the work to improve the English is appreciated.  Unfortunately, there are still a large number of inaccurate and unclear statements in section 2, Pathogenesis of PD.  This is section is not necessary for the review since is does not directly pertain to non-coding RNA. In my opinion, if section 2 is removed, the manuscript would be acceptable for publication.

Response: Thank you very much for your question and careful look. According to your suggestion, we removed the section 2, Pathogenesis of PD.

Reviewer 2 Report

N/A

Author Response

Thank you very much for your question and careful look. According to another reviewer’s opinion, we removed section 2, Pathogenesis of PD, in order to ensure the manuscript became more accurate and clear.